# The Liver and Lysosomal Storage Diseases: From Pathophysiology to Clinical Presentation, Diagnostics, and Treatment

**DOI:** 10.3390/diagnostics14121299

**Published:** 2024-06-19

**Authors:** Patryk Lipiński, Anna Tylki-Szymańska

**Affiliations:** 1Institute of Clinical Sciences, Maria Skłodowska-Curie Medical Academy, 00-136 Warsaw, Poland; 2Department of Pediatrics, Nutrition and Metabolic Diseases, The Children’s Memorial Health Institute, 04-730 Warsaw, Poland; a.tylki@ipczd.pl

**Keywords:** liver, hepatomegaly, splenomegaly, elevated transaminases, liver fibrosis, Gaucher disease, acid sphingomyelinase deficiency, lysosomal acid lipase deficiency, Niemann–Pick type C disease, enzyme replacement therapy, liver transplantation

## Abstract

The liver, given its role as the central metabolic organ, is involved in many inherited metabolic disorders, including lysosomal storage diseases (LSDs). The aim of this manuscript was to provide a comprehensive overview on liver involvement in LSDs, focusing on clinical manifestation and its pathomechanisms. Gaucher disease, acid sphingomyelinase deficiency, and lysosomal acid lipase deficiency were thoroughly reviewed, with hepatic manifestation being a dominant clinical phenotype. The natural history of liver disease in the above-mentioned lysosomal disorders was delineated. The importance of Niemann–Pick type C disease as a cause of cholestatic jaundice, preceding neurological manifestation, was also highlighted. Diagnostic methods and current therapeutic management of LSDs were also discussed in the context of liver involvement.

## 1. Introduction

The liver is an organ in which many biochemical pathways for the proper functioning of the body take place; that is why many inherited metabolic diseases manifest themselves particularly intensively in the liver (almost exclusively in the liver (hepatic glycogenoses) or mainly in the liver (urea cycle defects, sphingolipidoses)) [1,2]. Inherited metabolic diseases account for about one third of pediatric patients with hepatomegaly, cholestasis, acute liver failure or cirrhosis [3].

Lysosomal storage diseases (LSDs) are a group of inherited metabolic diseases associated with the dysfunction of lysosomal apparatus. More than half of LSDs are caused by a deficiency of lysosomal enzyme (mostly hydrolases and sulfatases) activity [4,5,6]. The accumulation of unmetabolized substrate plays a key role in the pathogenesis of an LSD. Lysosomal diseases may also be caused by defects in proteins involved in the intracellular transport, activator, and receptor functions in the lysosomal membrane, as well as disturbances in the post-translational modification of lysosomal enzymes (i.e., incorrect glycosylation) [7,8,9]. LSDs are inherited in an autosomal recessive manner, except for Fabry, Hunter (Mucopolysaccharidosis type II) and Danon diseases, which are X-linked recessive [4,5,6].

Lysosomal diseases are multi-systemic (enzyme deficiency or dysfunction of the lysosomal apparatus affects every cell type), but they could manifest themselves more intensely in selected tissues/organs, i.e., the monocyte–macrophage system in regard to Gaucher disease or nerve cells in gangliosidoses [1,4,5,6]. The severity of clinical consequences depends on the distribution of the substrate and its type (different in various types of cells and tissues).

LSDs manifesting early in life (neonatal/infantile period, childhood) have a more severe clinical course; such a manifestation could indicate a high dynamic/progression of the disease. Clinical models of LSDs were described based on early onset forms, but with the development of biochemical/molecular diagnostics, late-onset (usually milder) forms were identified [4,5,6].

The aim of this article was to provide a comprehensive overview on liver involvement in lysosomal storage diseases, focusing on clinical manifestation and its pathomechanism. Diagnostic methods and current therapeutic management of LSDs were also discussed in the context of liver involvement.

## 2. Liver in Lysosomal Storage Diseases: From Pathophysiology to Clinical Presentation

Most of the inherited metabolic diseases are characterized by liver cell (hepatocyte) involvement, which manifests itself as increased (usually slightly to moderately) transaminase (mainly alanine aminotransferase, ALT) activities. The exception concerns diseases involving the storage of macrophages of the reticuloendothelial system, which clinically manifest in the form of hepatosplenomegaly and constitute the dominant symptom in sphingolipidoses, especially Gaucher disease (GD; MIM # 230800), see Table 1 [1,10,11]. Moderate to severe hepatosplenomegaly with normal serum transaminases is observed in most patients with GD [10,11]. Hepatic involvement in GD very rarely leads to the end-stage liver disease and its complications, including portal hypertension [10,11].

In both acid sphingomyelinase deficiency (ASMD, formerly Niemann–Pick disease type A, B, A/B) and lysosomal acid lipase deficiency (LAL-D), an unmetabolized substrate accumulates in the macrophages and lysosomes of hepatocytes, which is manifested as hepatosplenomegaly with increased activity of transaminases (Table 1) [12,13,14]. In the natural history of liver disease in ASMD and LAL-D, progression to liver fibrosis is observed [12,13,14].

GD, ASMD and LAL-D represent lysosomal diseases with a dominant hepatic manifestation [14]. A special thought should be given to cholestatic jaundice in the neonatal/early infantile period (usually accompanied by hepatosplenomegaly), which may be the first symptom of Niemann–Pick type C disease (NPC) [15]. Neonatal cholestasis is not typical for GD, while it could be observed in early onset LAL-D as well as ASMD [13,16].

The largest group of lysosomal diseases involves those affecting the central nervous system (CNS), with the neurological decline occurring at various rates, while signs and symptoms related to the involvement of visceral organs (including hepatomegaly as a result of accumulation in the connective tissue stroma), although visible, do not dominate the clinical picture; this fact reflects all types of mucopolisaccharidoses, GM1 and GM2 gangliosidoses, mucolipidoses and glycoproteinoses (see Table 1) [1,4,5,6,14].

## 3. The Liver and Gaucher Disease, Acid Sphingomyelinase Deficiency and Lysosomal Acid Lipase Deficiency: Dominant Hepatosplenomegaly with or without Elevated Transaminases

### 3.1. Gaucher Disease

Gaucher disease (GD) is the most frequent sphingolipidosis, caused by biallelic pathogenic variants in the *GBA1* gene (MIM * 606463) encoding for β-glucocerebrosidase (GCase, E.C. 3.2.1.45) [11]. The storage of undegraded glucocerebroside (due to GCase deficiency) in Browicz–Kupffer cells is observed, which leads to hepatosplenomegaly without damage of hepatocytes [10,11]. Most symptoms of GD results from the involvement of cells of the monocyte–macrophage system; however, the criterion differentiating the individual types is the involvement of the central nervous system (CNS) [17]. Non-neuronopathic GD (type 1 GD; # 230800) is the most common form of the disease, defined by the constellation of visceral signs (hepatosplenomegaly, thrombocytopenia, bone disease), and the absence of primary CNS involvement [10,11]. The most common pathogenic genetic variant in *GBA1* in the Caucasian population is NM_000157.4:p.(Asn409Ser) [N370S], responsible for GD type 1 (GD1); the presence of N370S on one *GBA1* allele is protective of the development of a neurological involvement [18]. Most Polish GD1 patients are found to be heterozygous for N370S and other *GBA1* variants, especially L444P [c.1448T>C, p.(Leu483Pro)].

GD type 2 (# 230900) has been termed as acutely neuronopathic because of progressive neurodegeneration, while GD type 3 (# 231000), a subacute neuronopathic form, has a later onset of CNS signs and a highly variable rate of progression. Opisthotonus, bulbar signs (swallowing disorders), oculomotor paralysis (or bilateral fixed strabismus), and hepatosplenomegaly in the first 3–6 months of a patient’s life are very suggestive of GD type 2 [19]. GD3 patients usually present with similar somatic signs and symptoms to those observed in GD1 [20,21]. However, Polish GD type 3 patients (mostly L444P homozygotes) are characterized by an early onset (first 2 years of life) of massive hepatosplenomegaly [20,21]. Subtle neurological features, such as supranuclear gaze palsy and a mask-like face, generally appear later. This characteristic is very similar to the Norrbottnian-derived Swedish Gaucher population [21].

Since hepato-splenomegaly and thrombocytopenia are the dominant clinical features in GD patients, pediatricians and pediatric gastroenterologists need to be aware of this condition [10,11]. Hepatic involvement in GD rarely progresses to end-stage liver disease [14]. However, liver fibrosis in GD seems to be underestimated based on the results of non-invasive techniques such as ultrasound (US)-based elastography or magnetic resonance (MR) elastography. Splenectomy and non-N370S *GBA1* genotypes are well-known factors associated with the presence and severity of liver fibrosis in GD [22,23,24]. Diffuse steatosis in GD occurs in adult patients as a part of metabolic syndrome either de novo or as a side effect of long-term enzyme replacement therapy (ERT) [25].

### 3.2. Acid Sphingomyelinase Deficiency

Acid sphingomyelinase deficiency (ASMD) is caused by biallelic pathogenic variants in the *SMPD1* gene (* 607608). As a result, an undegraded sphingomyelin accumulates in both the macrophages and lysosomes of hepatocytes, leading to (moderate to severe) hepatosplenomegaly and increased activity of transaminases [14,26,27,28]. Sphingomyelin is primarily the main component of the myelin sheath, hence its accumulation within neurons and symptoms of CNS involvement are observed in neuronopathic forms of the disease. The infantile neurovisceral form (Niemann–Pick disease type A; # 257200) is characterized by the rapidly progressive involvement of visceral organs (hepatosplenomegaly—may be massive, moderately increased activity of transaminases, possible cholestatic jaundice) and neurodegeneration, leading to death before the age of 3 years [26,27,28]. Neurological manifestation is characterized by a progressive psychomotor regression followed by extrapyramidal symptoms and increasing spasticity. Patients with chronic visceral disease (Niemann–Pick type B disease; NPB; # 607616) have different ages at the onset of symptoms (from childhood to adulthood) and slowly progressing symptoms related to the involvement of visceral organs without CNS involvement [26,27,28]. The clinical classification also distinguishes an intermediate phenotype (type A/B), which is a chronic neurovisceral form of the disease with a later onset of first symptoms than in the infantile form, slower progression of CNS and visceral organ involvement, and longer survival rates [26,27,28,29].

Moderate to severe liver enlargement is a permanent feature of chronic visceral ASMD, while splenomegaly is present in almost all patients [30,31,32]. In a cross-sectional (United States, Brazil, Italy, France, Germany) study of 59 patients (median age 17.6 years) with chronic visceral ASMD, ALT was elevated in 51% of them [30]. In a natural history study (Mount Sinai General Clinical Research Center) of 29 patients (aged 2–64 years of age at study entry), 75% of them had elevated ALT at the initial visit [31]. No statistically significant changes in liver enzyme values were noted during follow-up (mean: 4.3 years) [31]. In a Polish cohort of patients, an elevated ALT was observed in 38% of patients at diagnosis and remained comparable during follow-up (mean: 10 years) [32].

Most patients with chronic visceral ASMD disease show also radiological signs of lung involvement (so-called “ground-glass” image); however, no correlation between radiological and clinical features has been demonstrated [26,27,28,30,31,32]. A typical lipid serum profile is also observed in most patients, namely low HDL cholesterol and elevated LDL cholesterol and triglycerides.

There are several reports of liver fibrosis and cirrhosis in Niemann–Pick type B patients [12,33,34]. One of the potential mechanisms is that the sphingomyelin storage in lysosomes could provoke a fibrotic reaction in tissues. McGovern et al. studied morbidity and mortality in a cohort of 103 patients ranging from 1 to 72 years of age [12]. Nine patients had cirrhosis or liver failure, requiring liver transplantation (LTx). Of note, six patients had fulminant liver failure and three of them had evidence of cirrhosis upon liver biopsy. Cassiman et al. characterized disease-related morbidities and causes of death in patients with chronic visceral and chronic neurovisceral ASMD [34]. The overall leading causes of death were respiratory failure and liver failure (27.7%), irrespective of age. Death due to liver disease was equally frequent in pediatric and adult patients. In contrast, our results showed that chronic visceral ASMD could constitute a slowly progressing disease with a relatively good outcome [32].

### 3.3. Lysosomal Acid Lipase Deficiency

Lysosomal acid lipase (LAL) deficiency is caused by biallelic pathogenic variants in the *LIPA* gene (* 613497). In addition to the storage of cholesteryl esters and triglycerides within macrophages of the reticuloendothelial system, hepatocytes are also damaged in LAL-D [35]. Early onset LAL-D (known also as Wolman’s disease; # 620151) is characterized by its onset in the first weeks of life. Characteristic features include a significant and progressive enlargement of the liver and spleen, vomiting and diarrhea (the result of malabsorption due to the accumulation of lipids in the intestinal mucosa) and failure to thrive [13,35,36,37]. Enlargement and calcification of the adrenal glands are characteristic but not pathognomonic symptoms, and they do not occur in all patients. Anemia and thrombocytopenia (due to progressive hypersplenism) as well as malnutrition intensify as the disease progresses. Other biochemical abnormalities include increased activity of transaminases (also gamma-glutamyltranspeptidase) and coagulopathy (hypofibrinogenemia, prolonged INR, hypoalbuminemia) [13,35,36,37]. In some patients there were observed features of the secondary hemophagocytic syndrome—hyperferritinemia, hypertriglyceridemia and hemophagocytosis in bone marrow smears [36]. The disease is characterized by rapidly progressive cachexia, liver, and adrenal cortex failure, which lead to death in infancy, usually before the age of 4 months [37].

Late-onset LAL-D, also called cholesteryl ester storage disease (CESD; # 278000), is characterized by a mild onset in the first or second decade of life or even later. The most characteristic feature is moderate liver enlargement, which may persist for many years before the diagnosis [38,39,40,41,42]. The main biochemical abnormality is hyperlipidemia, defined as increased concentrations of total cholesterol, triglycerides, and LDL cholesterol, as well as normal/decreased HDL cholesterol [38,39,40,41,42]. In a majority of patients, a mild to moderate increase in serum transaminases activities is observed [38,39,40,41,42]. Liver ultrasounds usually show signs of hepatic steatosis, while the histopathological examination of a liver biopsy reveals microvesicular steatosis and foamy macrophages (in almost all patients). Pathognomonic birefringent crystals could be observed in hepatocytes and Kupffer cells under polarized light [14,38]. Liver damage may progress, leading to liver fibrosis and, ultimately, cirrhosis [38,39,40,41,42].

In a group of 19 Polish patients with late-onset LAL-D under the care of our institute, liver enlargement and hyperlipidemia were observed in all of them, while increased ALT was found in 79% of patients [38]. The average age of patients at diagnosis was 7 years and 2 months. A liver biopsy was performed in 15 out of all 19 patients at the mean age of 6 years and 6 months. The evidence of fibrosis was reported in 7 (47%) patients and cirrhosis in 3 (20%) patients [38].

According to the latest (May 2023) data from the international (23 countries, including Poland) registry of patients with LAL-D (International LAL-D Registry), out of 252 total patients, 228 (139 children and 89 adults) were patients with late-onset LAL-D [42]. This is the largest cohort of patients with late-onset LAL-D that has been reported to date. The average age of pediatric patients with late LAL-D at the time of first complaints and symptoms was 3.8 years vs. 7.1 years at the time of diagnosis. Liver enlargement was observed in 77% of children and 50% of adults, while spleen enlargement was observed in 37% of patients (equal percentage in children and adults). The most common abnormalities in laboratory tests were increased ALT activity (75% of children) and increased LDL cholesterol (60% of children). Among 118 patients with liver biopsies, 63% had microvesicular steatosis exclusively, 23% had mixed micro- and macrovesicular steatosis and 47% had lobular inflammation. Of 78 patients with fibrosis-stage data, 37% had bridging fibrosis and 14% had cirrhosis.

## 4. Liver in Niemann–Pick Type C Disease—Hepato/Splenomegaly with Cholestatic Jaundice

### Niemann–Pick Type C Disease

Niemann–Pick type C disease (# 257220, # 607625; NPC) is a lysosomal storage neurodegenerative disease caused by mutations in either the *NPC1* or *NPC2* gene (* 607623, * 601015) coding for proteins, working in a coordinated manner, involved in the cellular trafficking of cholesterol and other lipids in the late endosomal/post-lysosomal stage [43].

About half of patients with NPC present in the first year of life with cholestatic jaundice and hepatosplenomegaly. Although jaundice disappears within several months in most patients, splenomegaly remains [15,44].

The congenital form of NPC is a rare form of the disease; however, the main clinical feature is liver involvement associated with hepatocyte damage, unlike in other neurological forms in which liver involvement is mild and mainly associated with macrophage involvement [45]. Prenatally, fetal edema, ascites and polyhydramnios may be observed, while, postnatally, hepatosplenomegaly and rapid progression of the disease leading to liver failure and death may be observed.

Disease classification is based on the age of onset of the first (beyond 3 months of life) neurological symptoms; thus, the following forms are distinguished:Early infantile form: Involvement of parenchymal organs precedes neurological manifestation (onset of symptoms between 2 months and 2 years of age). Splenomegaly or hepatosplenomegaly (possible from birth) and prolonged cholestatic jaundice are characteristic [15,43,46,47]. Delayed psychomotor development and axial hypotonia are usually the first neurological symptoms. The subsequent clinical course includes loss of acquired motor skills, followed by marked spasticity and other pyramidal symptoms.Late-infantile form (first noticeable neurological symptoms 3–6 years of age) and juvenile form (6–15 years of age): Features a history of prolonged cholestatic jaundice, isolated splenomegaly or hepatosplenomegaly in infancy, which may also persist from infancy onwards [43,46,47]. Neurological symptoms in the infantile form include gait disturbances, ataxia, vertical supranuclear gas palsy, intellectual disability, and cataplexy. A significant proportion of patients develop epileptic seizures. As the disease progresses, dysphagia, dysarthria, pyramidal symptoms, and spasticity occur. The juvenile form is the most common form of NPC in most countries. The first symptoms are usually learning difficulties, writing problems, and upward gaze paralysis. The neurological manifestation of the juvenile form also includes ataxia, dysarthria, dysphagia, dystonia, pyramidal symptoms, and spasticity [43,46,47].Adult form (>16 years of age): Psychotic symptoms in at least 1/3 of patients in the form of psychosis and depression. Neurologic manifestations include ataxia, upward gaze paralysis, dysarthria, dysphagia, and dystonia. In this form, there is no enlargement of the spleen or liver [43,48].

The phenotype of liver involvement in Gaucher disease, acid sphingomyelinase deficiency, lysosomal acid lipase deficiency and Niemann–Pick type C disease is summarized in Table 2.

## 5. Diagnostics of Lysosomal Diseases

The gold standard of diagnostics of lysosomal diseases associated with enzyme deficiency (GD, ASMD, LAL-D) is the demonstration of deficient enzyme activity (β-glucocerobrosidase, acid sphingomyelinase, lysosomal acid lipase, respectively) in peripheral blood leukocytes or cultured skin fibroblasts (Table 2) [49]. Recently, the enzyme activity could be measured in dried blood spot (DBS) samples as the first-line laboratory screening method. The identification of biallelic pathogenetic variants in the corresponding genes (*GBA1*, *SMPD1*, *LIPA*, respectively) confirms the biochemical diagnosis (Table 2).

The use of dried blood spots (DBSs) in the diagnostics process of LSDs has become increasingly popular, mainly due to its convenience. In addition to enzyme activity measurement, simultaneous diagnostics based on substrate identification is possible, i.e., analysis of glucosphingosine (lyso-Gb1) concentration in GD and lyso-sphingomyelin (lyso-SPM) in ASMD [50,51,52]. Once NPC is suspected clinically, several metabolites (cholestane-3β, 5α, 6β-triol, lyso-sphingomyelin isoforms—lyso-SM and lyso-SM509—and bile acid metabolites) have emerged as sensitive and specific diagnostic biomarkers for NPC and their study, completed by genetic analyses (*NPC1* and *NPC2* genes), and they should now be considered as the first line in laboratory testing [53]. Clinical applications of the above-mentioned biomarkers are found in improved diagnostics, monitoring disease progression and assessing therapeutic efficacy.

In every case of hepatosplenomegaly of an unknown cause, we acknowledge the need to perform a chitotriosidase activity assessment. Chitotriosidase (E.C. 3.2.1.14) is an enzyme produced and secreted in large amounts by activated macrophages, especially macrophages loaded with phagocytized glycosphingolipid in GD, and, to a lesser extent, in other lysosomal diseases, including LAL-D, ASMD and NPC [14]. Regarding GD, it has been used for more than 20 years as a biomarker reflecting the disease severity as well as treatment efficacy [18]. Chitotriosidase activity assessments are still performed in many centres, proving their usefulness.

## 6. Treatment of Lysosomal Diseases Regarding Liver Involvement

### 6.1. Enzyme Replacement and Substrate Reduction Therapies: Overall Issues

Enzyme replacement therapy (ERT) is currently the gold standard for the treatment of LSDs, and its efficacy was proven for the first time in GD patients [54]. The recombinant enzyme (imiglucerase or velaglucerase alfa for GD, sebelipase alfa for LAL-D, olipudase alfa for ASMD) is administered intravenously and then taken up by target tissues, mainly macrophages of the reticuloendothelial system, via the mannose-6-phosphate receptor [54]. ERT effectively reduces liver and spleen volume in GD, ASMD, and LAL-D patients [55,56,57,58].

Substrate reduction therapy (SRT) constitutes the other treatment option in some LSDs and relies on the small-molecule inhibitors of enzymes involved in the biosynthesis of substrates [59]. Miglustat was the first approved SRT dedicated to adult GD1 patients with mild-to-moderate disease in whom ERT was either unsuitable or was not a therapeutic option. It has been shown to be effective in improving or stabilizing visceral (reducing liver and spleen volume), hematologic and bone markers of the disease. On the other hand, Miglustat is approved for the treatment of neurological features of NPC. It has been shown to stabilize neurological disease progression in pediatric patients with NPC with comparable safety and tolerability to that observed in adults and juveniles [60].

The approved (2015 year) SRT for GD is currently eliglustat. By reducing substrate influx, eliglustat has significantly improved disease manifestations in patients with existing visceral and hematologic involvement [61].

Both ERT and SRT have demonstrated being highly effective in reducing the liver volume of GD patients [10,55,61].

In patients with ASMD treated with olipudase alfa, improvement in liver function tests and plasma lipid profiles was observed at 12 months and was maintained or further improved at 24 months, with mean values being maintained within normal ranges [56,57].

Sebelipase alfa reduced serum aminotransferases and other markers of liver cell injury in LAL-D patients. At week 52, mean alanine aminotransferase and aspartate aminotransferase levels were found to be normal [58]. In the pivotal clinical trial, after 52 weeks, sebelipase-alfa prevented or reversed fibrosis and cirrhosis in 11 of 12 biopsied patients, and no long-term ERT patient progressed to requiring liver transplantation (LTx) [58].

### 6.2. Symptomatic (Liver-Related Symptoms) Treatment

In the era of ERT, splenectomy should be avoided in GD patients [10]. This statement also involves patients with ASMD for whom ERT was recently available [27]. The removal of the spleen eliminates an important reservoir for substrate accumulation that leads to acceleration of disease in other organs.

Liver transplantation (LTx) may be proposed for the rare patients presenting with severe liver disease progressing to liver failure. LTx in GD patients is very rarely reported [10,62]. Similarly, the literature contains only few reports of patients with chronic ASMD undergoing LTx [63,64].

Liu et al. evaluated the effects of LTx in seven children with NPB (single-center experience) [64]. The median age at diagnosis was 12 months (6–14 months) with initial presentations of hepatosplenomegaly, growth retardation, repeated pneumonia, and diarrhea. All patients developed severe liver dysfunction, interstitial pulmonary disease, compromised lung function, and hypersplenism. LTx was performed at a median age of 6.5 years (range 2.2–8.6 years). All patients were alive with a median follow-up of 10 months (range 5–53 months). Liver function normalized within 3 weeks after LTx and maintained stability. Strikingly, pulmonary disease was relieved after LTx, as evidenced by resolution of interstitial lung disease and restored lung function.

LTx may be also requisite for LAL-D liver failure but does not prevent systemic LAL-D progression. Bernstein et al. analyzed 18 cases of LTx in a group of patients with LAL-D [65]. The mean age of patients at the time of LTx was 12 years (range: 7 months–42 years). Progression of the underlying disease and damage to the transplanted liver were observed in 11 patients (61%), while death was observed in 6 patients (33%).

The research on effective treatment of NPC is still ongoing, and LTx still remains controversial in regard to NPC [66]. Modin et al. reported nine children with NPC with neonatal (median age at first presentation: 7 days) acute liver failure for whom LTx was performed [66]. However, it needs to be highlighted that NPC diagnosis was made after LTX in eight out of nine patients. Seven children were still alive (a median follow-up time of 9 (range: 6–13) years), while neurological symptoms developed in four out of seven patients at a median of 9 (range: 5–13) years following LTx [67]. Yamada et al. reported three patients with neonatal-onset NPC in whom LTx was performed as a life-saving procedure. In two patients, the diagnosis of NPC was made more than a year after LTx [68]. All patients showed neurological regression and required artificial respiratory support. In conclusion, while LTx may be a temporary life-saving procedure in patients with neonatal-onset NPC and liver failure, the outcome is poor, especially due to neurological decline.

### 6.3. Liver-Targeted Gene Therapy

Gene therapy targeting the liver has become an attractive therapy for monogenic disorders [69]. From a pathophysiological point of view, it seems to be a promising option for LAL-D, which has a predominant liver phenotype. Lam et al. reported that rscAAVrh74.miniCMV.LIPA gene therapy significantly improved the disease symptoms in the Lipa^−/−^ mouse model of LAL-D, namely lowered hepatosplenomegaly and significantly reduced transaminases, as well as reduced liver and spleen triglyceride and cholesterol levels [70]. Notably, rscAAVrh74.miniCMV.LIPA gene therapy did not significantly elevate serum transaminase levels at any of the time points tested, indicating that adeno-associated virus (AAV) treatment does not contribute to liver damage. Laurent et al. have recently published the results of in vivo gene therapy using recombinant AAV8 and a stable long-term LAL expression sufficient for correcting the disease phenotype in the Lipa^−/−^ mouse model [71]. Expressed LAL ameliorated mouse weight and growth rate, corrected hepatosplenomegaly, dyslipidaemia and transaminase, and it also reduced lipid accumulation in the jejunum.

The development of gene therapy for GD has focused on AAV vectors that allow for achieving a stable expression of target genes in the nervous system. Glucocerebrosidase activity was significantly increased in the brain, liver, and spleen, and no toxic effects were detected [72]. While the stable expression in the nervous system is the most important factor for treating neuronopathic GD, it is also desirable to express GBA1 systemically to treat pathology in visceral organs [72]. Currently, there are a few active gene therapy clinical trials for the treatment of GD [73]. Miranda et al. reported promising results regarding liver-directed gene therapy in vitro and in vivo for the treatment of GD. Liver-directed GBA AAV vector administration resulted in a sustained elevation of glucocerebrosidase in the bloodstream and a higher level of its bioavailability for uptake into macrophages than with ERT (velaglucerase alfa) [74].

## Figures and Tables

**Table 1 diagnostics-14-01299-t001:** Lysosomal storage diseases with enlargement of the liver and spleen.

Sphingolipidoses	Gaucher’s disease, acid sphingomyelinase deficiency (Niemann–Pick disease A, B, A/B), infantile form of GM1-gangliosidosis, GM2-ganliosidosis (Sandhoff’s disease)
Mucopolysacchardoses (MPS)	MPS I, II, VI, VII
Lysosomal lipid storage	Niemann–Pick disease type C, lysosomal acid lipase deficiency (Wolman’s disease and cholesteryl ester storage disease)
Mucolipidoses (ML)	ML I (sialidosis)
Glycoproteinoses	α-mannosidosis, galactosialidosis, type II glycogenosis (Pompe disease)

**Table 2 diagnostics-14-01299-t002:** Clinical and biochemical features of liver expression in selected lysosomal diseases with diagnostics methods (“+”; present).

Disease	Liver Enlargement	Increased Transaminases Activity	Cholestasis	Acute Liver Failure	Liver Steatosis	Liver Fibrosis/Cirrhosis	Liver Cancer	Diagnostics
Gaucher disease (GD)	+					+ Rarely	HCC casuistically	β-glucocerebrosidase activity in peripheral blood leukocytes/skin fibroblasts/dried blood spot (DBS); Lyso-Gb1 in DBS; Chitotriosidase in blood serum or DBS; Molecular testing (*GBA1* gene).
ASMD	+	+	+ Infantile type			+ Casuistically		Acid sphingomyelinase activity in peripheral blood leukocytes/skin fibroblasts/DBS; Lysosphingolipids, i.e., lyso-SM and lyso-SM-509 in DBS, determined by LC-MS/MS, Chitotriosidase in blood serum; Molecular testing (*SMPD1* gene)
LAL deficiency (LAL-D)	+	+	+ Wolman disease		+	+		Lysosomal lipase (LAL) activity in peripheral blood leukocytes/skin fibroblasts/DBS; Chitotriosidase in blood serum; Molecular testing (*LIPA* gene).
Niemann–Pick type C disease (NPC)	+	+	+	+ Congenital type		+	HCC	Lysosphingolipids, i.e., lyso-SM and lyso-SM-509 in DBS, determined by LC-MS/MS, and oxysterols (cholestane-3β, 5α, 6β-triol, 7-ketocholesterol); Chitotriosidase in blood serum; Molecular testing (*NPC1*, *NPC2* genes).

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
