# Peer review of "The Liver and Lysosomal Storage Diseases: From Pathophysiology to Clinical Presentation, Diagnostics, and Treatment"

_diagnostics, 2024, doi:10.3390/diagnostics14121299_

Round 1

Reviewer 1 Report

Comments and Suggestions for Authors

The manuscript is well written and clearly summarize the current knowledge and evidence of the involvment of liver in LSDs. This review in its current form is highly acceptable for publication.

Author Response

Dear Reviewer

We are very grateful for Your kind revision.

Best wishes

Patryk Lipiński

Reviewer 2 Report

Comments and Suggestions for Authors

In the review by Lipinski et al. the authors present a description of liver pathology in different lysosomal diseases. This focus is interesting, however the article can be improved with a few more information. 

In the description of Polish GD patients the authors state that the patients are heterozygous for the N370S mutations. Since GD is a recessive disease, if these patients are symptomatic, there is a high probability that they present the N370S on one allele and an unidentified mutation on the other allele (ej. large deletion, deep intronic, duplication). The author should state that the second mutation was not identified in this cohort.

In the description of the treatment it is stated that Miglustat is approved for GD, however this treatment is sporadically prescribed for GD off-label and the approved SRT for GD is currently Eliglustat.

On the other hand, Miglustat is approved for the treatment of NPC, but it was not mentioned in the article (Patterson MC, Vecchio et al. Child Neurol 2010; 25:300-5).

As a final comment it would enrich the significance of the review to comment on possible effects of the liver-directed gene therapy approaches that are under development (GD y LAL) on the pre-existing liver pathology. 

Minor points: 

Please add the ORCID codes and the gene IDs when introducing each of the lysosomal disorders. 

Author Response

In the review by Lipinski et al. the authors present a description of liver pathology in different lysosomal diseases. This focus is interesting, however the article can be improved with a few more information. 

Answer:

Dear Reviewer, we are very grateful for Your kind review. 

In the description of Polish GD patients the authors state that the patients are heterozygous for the N370S mutations. Since GD is a recessive disease, if these patients are symptomatic, there is a high probability that they present the N370S on one allele and an unidentified mutation on the other allele (ej. large deletion, deep intronic, duplication). The author should state that the second mutation was not identified in this cohort.

Answer:

Most Polish GD1 patients are found to be heterozygous for N370S and other GBA1 variant, especially L444P [c.1448T>C, p.(Leu483Pro)].

In the description of the treatment it is stated that Miglustat is approved for GD, however this treatment is sporadically prescribed for GD off-label and the approved SRT for GD is currently Eliglustat.On the other hand, Miglustat is approved for the treatment of NPC, but it was not mentioned in the article (Patterson MC, Vecchio et al. Child Neurol 2010; 25:300-5).

Answer: It was corrected as advised.

The following text was added/revised: 

On the other hand, Miglustat is approved for the treatment of neurological features of NPC. It was shown to stabilize neurological disease progression in pediatric patients with NPC with comparable safety and tolerability to that observed in adults and juveniles [60].

The approved (2015 year) SRT for GD is currently Eliglustat. 

As a final comment it would enrich the significance of the review to comment on possible effects of the liver-directed gene therapy approaches that are under development (GD y LAL) on the pre-existing liver pathology. 

Answer: It was corrected as advised.

The following paragraph was added:

6.3 Liver-targeted gene therapy

Gene therapy targeting the liver has become an attractive therapy for monogenic disorders [69]. From pathophysiological point of view, it seems to be a promising option for LAL-D, which has a predominant liver phenotype. Lam et al. reported that rscAAVrh74.miniCMV.LIPA gene therapy significantly improved the disease symptoms in the Lipa-/- mouse model of LAL-D [70]. Laurent et al. have recently published the results of in vivo gene therapy using recombinant AAV8 and a stable long-term LAL expression sufficient to correct the disease phenotype in the Lipa-/-mouse model.

While the stable expression of target genes in the nervous system is the most important factor for the development of adeno-associated virus (AAV) to treat neuronopathic GD, it is also desirable to express GBA1 systemically to treat pathology in visceral organs [71]. Currently, there are a few active gene therapy clinical trials for the treatment of GD [72]. Miranda et al. reported a promising results of liver-directed gene therapy in vitro and in vivo for the treatment of GD. Liver directed GBA AAV vector administration resulted in a sustained elevation of glucocerebrosidase in the bloodstream and higher level of its bioavailability for uptake into macrophages than ERT (velaglucerase alfa) [73].

Minor points: 

Please add the ORCID codes and the gene IDs when introducing each of the lysosomal disorders. 

Answer: It was corrected as advised.

Round 2

Reviewer 2 Report

Comments and Suggestions for Authors

Thank you for the revision. However, regarding the gene therpay I ment that the author should comment if the transgene expression in the liver may or not affect the liver pathology, since AAV expression may transitorially increase transaminases. Do clinical trials for GD or LAL report any toxicity for the liver?

Author Response

Dear Reviewer

Thanks for Your comments. 

AAV gene therapy in both GD and LAL-D did not elevate serum transaminase levels, indicating that AAV treatment itself is not contributing to liver damage.

This paragraph was revised, please find attached the corrected version.

Kind regards

Patryk Lipiński
